# SemG-TS: Abstractive Arabic Text Summarization Using Semantic Graph Embedding

**Wael Etaiwi \*** and **Arafat Awajan**

Princess Sumaya University for Technology, Amman 11941, Jordan
\* Correspondence: w.etaiwi@psut.edu.jo

**Abstract:** This study proposes a novel semantic graph embedding-based abstractive text summarization technique for the Arabic language, namely SemG-TS. SemG-TS employs a deep neural network to produce the abstractive summary. A set of experiments were conducted to evaluate the performance of SemG-TS and to compare the results to those of a popular baseline word embedding technique called word2vec. A new dataset was collected for the experiments. Two evaluation methodologies were followed in the experiments: automatic and human evaluations. The Rouge evaluation measure was used for the automatic evaluation, while for the human evaluation, Arabic native speakers were tasked to evaluate the relevancy, similarity, readability, and overall satisfaction of the generated summaries. The obtained results prove the superiority of SemG-TS.

**Keywords:** abstractive text summarization; semantic graph; semantic graph embedding; Arabic text summarization

**MSC:** 68T50; 68T07

## 1. Introduction

Due to the rapid increase in the number of electronic documents, articles, and pages on the Internet, the need to summarize their content has emerged [1]. When online content rises at a fast pace, finding relevant information becomes a more difficult mission. Users can get distracted and thus miss catching and reading valuable and interesting material. There is therefore a need for a text summarization solution. Text summarization compresses a large volume of texts from various sources (such as documents, web sites, and comments) into a shorter length and concise summary [2,3]. The automatic extraction or creation of a summary of a given text is called text summarization. Several challenges have been identified while summarizing documents [4], such as: (1) Redundancy, which can lead to the final summary including redundant information. (2) Irrelevancy, in which the final summary may contain irrelevant information. (3) Coverage loss, in which a key detail is missed in the final summary. In addition, (4) the final summary may not be readable if it comprises unrelated words. Text summarization could be categorized into many ways and according to many factors [1].

- According to the number of documents, text summarization is categorized into single or multi-document summarization. The task of multi-document summarization is more difficult and has many additional challenges and issues that should be considered and solved, such as content redundancy.
- Text summarization is categorized into two main types based on the output summary: extractive and abstractive. In extractive text summarization, the summary is generated by ranking and selecting the most relevant text components (such as sentences) from the original text. In abstractive text summarization, the summary is produced from scratch, including words and expressions that may not exist in the original text.

Therefore, an abstractive summary preserves the main ideas in the original text and re-interprets them into a different form by using varying words and phrases. Abstractive text summarization is much more sophisticated than extractive text summarization since it needs to employ extensive Natural Language Processing (NLP) processes.

- Based on the total number of statements in an output summary, text summarization is divided into two major groups. When the final summary comprises a single sentence at most, it is considered a single-statement summarization. Otherwise, it is called a multi-statement summarization.

The summarization of English texts has been the subject of several studies in the field of text summarization. However, challenges with text summarization have highlighted the need for more studies in order to increase the efficacy of current text summary techniques for languages other than English. For instance, according to Al Saleh, summarizing Arabic texts is more difficult than summarizing English texts. As a result, Arabic text summarizing techniques have not made as much progress as those used for other languages due to Arabic's distinctive characteristics. Few studies have been proposed to produce Arabic text summary methods [5]. This is primarily due to the Arabic language's complexity both in terms of syntax and morphology, the Arabic diglossia, the language's high levels of ambiguity, and its highly derivational and inflectional nature. Many of the proposed solutions for text summarization have concentrated on extracting text summarization rather than abstracting text summarization; this is because extractive text summarization is much simpler than abstractive text summarization [6]. In an extractive text summary, the extracted summary includes the most relevant statements in the original text, which may be long, complex, and difficult to understand. Although abstractive text summarization is more complex and challenging than extractive text summarization, it is needed to produce simple and human-friendly statements that describe the most important ideas of the original text.

Semantic representation is used by a number of Natural Language Processing (NLP) applications to improve outcomes in the field of computer linguistics (e.g., machine translation and question answering). The main goal of semantic representation is to create detailed notations of the text that accurately convey its meaning. Huge and complicated data structures are represented and formalized in a standard and formal fashion using graphs. Compared with other text representation schemes such as predicate logic representation, frame representation, and rule-based representation, the graph model is more efficient because it is characterized by its ability to represent the semantic relations between the words in a text [7].

The method of semantic representation that has been used most frequently is the semantic graph [8–11]. A semantic graph is a network that reflects the semantic relationship between different concepts (e.g., terms, and sentences). Graph vertices are concepts, while graph edges are semantic relationships between concepts. The semantic graph is used to encode plain text and represent its context as a graph. Semantic preservation is a difficult task in semantic graph representation because semantic relationships differ depending on the language of the text, and because they are hard to capture in some languages.

Text summarization employs several text features for sentences, paragraphs, and words. Traditional methods require the use of hand-crafted features [12], which takes time and effort to manually extract the useful features. However, deep learning makes it possible to generate useful features from training data. Instead of using hand-crafted features, which mostly rely on the prior expertise of designers and are extremely challenging to utilize with a massive amount of data, deep learning automatically learns features from the existing data.

Owing to the aforementioned challenges, a new framework for abstractive Arabic text summarization using Arabic semantic graph representation is required. The proposed framework is called SemG-TS. The morphological and syntactical characteristics of the Arabic language should be taken into consideration when creating the semantic graph and when learning it. Thus, the SemanticGraph2Vec model is used to preserve the semantic relationships between words in the text graph. Since deep learning has had positive

results in several different fields of AI and in data mining problems [13], it is used to produce a text summary output. Finally, in order to evaluate the proposed SemG-TS, a new text summarization dataset is created based on well-written and published news articles. Furthermore, two evaluation methodologies are followed in the experiments: automatic and human evaluations.

The remainder of this paper is organized as follows. In Section 2, a brief review of the related work on text summarization for the Arabic language is presented. The proposed model is described in Section 3. The dataset used in the experiments is described in Section 4. In Section 5, the experiments and evaluation results are discussed. The conclusions are summarized in Section 6.

## 2. Related Work

Text summarization is more than fifty years old; the research community is very active in this area [1]. Researchers continue to improve the performance of current text summarization approaches or propose novel summarization approaches to enhance the quality of the output summary. However, the output of current text summarization models is still at a moderate level.

Due to the Arabic language's intrinsic complexity, both in terms of structure and morphology, methods and approaches for summarizing Arabic texts are still immature and insufficient [5,14]. Arabic text summarization approaches are classified into three main groups: graph-based approaches, deep learning-based approaches, and genetic algorithm and machine learning-based approaches.

- Graph-based approaches: Belkebir and Guessoum [15] used a multi-graph to decompose the original text into a set of sentence subsets. The Bell Numbers Theory was used to calculate the number of subsets. Graph vertices represented sentences, while the edges between the sentences represented the relationships between the sentences. Each layer of the multi-graph represented a semantic relationship between the sentences of the document. A machine learning approach was used to pick the most appropriate (highly probable) operation for each layer (partition). The list of operations used in the experiments included: sentence extraction using the AdaBoost machine learning technique, concept generalization and fusion for abstractive sentence generation, and sentence compression. The TALAA-ASC corpus [16] was used to evaluate the proposed model in terms of precision, recall, F-score, and ROUGE. In another study, an Arabic text summarization approach for a single document in Arabic was proposed by Azmi and Altmami [17]. The proposed approach relied on an extractive approach that produced the highly ranked sentences to be included in the final summary. Subsequently, a rule-based reduction technique was used to reduce the size of the extracted sentences and to reshape their structure. The authors referred to this approach as an abstractive text summarization approach, which is not accurate. The abstractive summary should contain new terms that do not exist in the original text, and this was not applied in the proposed approach. The proposed approach was evaluated on a set of 150 news articles. The results were analyzed manually by two human experts who scored the output summary out of a maximum score of five. The average score was between 4.53 and 1.92.

  Elbarougy et al. [18] proposed an extractive graph-based Arabic text summarization technique. The proposed model represented the original text as a graph, where the vertices of the graph were the sentences. Each sentence (vertex) was initially ranked by the total number of nouns in the sentence. The weights of the graph edges were calculated using the cosine similarity between the sentences. The proposed model consisted of three main stages: The first stage was the pre-processing stage, which included normalization, tokenization, stop word removal, stemming, and morphological analysis. The next stage was the feature extraction and graph construction stage, in which the graph was constructed and features were extracted. The last stage was the application of the Modified PageRank algorithm and the extraction of the final

summary. The PageRank algorithm was used with a different number of iterations in order to get the number of iterations that would produce the best results. The Essex Arabic Summaries Corpus (EASC) containing 153 documents was used to evaluate the proposed model in terms of precision, recall, and F-score.

In another study, Elbarougy et al. [19] proposed a graph-based extractive Arabic text summarization approach using multi-morphological analysis. This proposed approach transformed the original text into a graph. The sentences were represented as vertices, and the relationships between the sentences were calculated using the cosine similarity between the sentences based on Term Frequency–Inverse Document Frequency (TF-IDF) and the mutual nouns between the connected sentences. Three morphological analyzer algorithms were used to improve the efficiency of the proposed text summarization approach: Buckwalter Arabic Morphological Analyzer (BAMA) [20], Safar Alkhalil [21], and Stanford Natural Language Processing (NLP) [22]. The experimental results of the Essex Arabic Summaries Corpus (EASC) showed that the Safar Alkhalil morphological analyzer performed better than the other three analyzers.

- Deep learning-based approaches: Alami et al. [23] proposed a new extractive Arabic text summarization method. Auto-encoder models were used by the authors to learn a feature space from high-dimensional input data. Many inputs were discussed in the proposed research, including: term frequency, local vocabulary, and global vocabulary. The input sentences were ranked on the basis of the representation provided by the auto-encoder model. Two description methods were used in the proposed model to study the impact of the auto-encoder model: graph-based text summarization and query-based text summarization. Two separate datasets were used in the experiments: the EASC and the authors' dataset, which included 42 news articles. The authors concluded that the auto-encoder using the Term Frequency–Inverse Document Frequency (TF-IDF) representation of global vocabularies provides a more discriminative feature space and increases the recall of other models for both graph-based and query-based summarization approaches.

  Qaroush et al. [24] proposed an extractive single document summary approach aimed at optimizing content coverage and consistency between sentences in the summary. The proposed approach satisfied the two opposing semantic goals of coverage and diversity by evaluating each sentence based on a combination of the most informative statistics and semantic properties. Two text summarization techniques were used to determine the appropriateness of the statistical and semantic features: score-based and supervised machine learning. The EASC dataset was used in the experiments in order to demonstrate the effectiveness of the proposed method. The proposed approach was domain-independent and did not include any domain-specific information or features. The experimental results showed the efficacy of the proposed approach in terms of precision, recall, and F-score.

- Genetic algorithms and machine learning-based approaches: Belkebir and Guessoum [25] turned the task of summarizing Arabic text into a process of prediction. They used a machine learning-based approach (called AdaBoost) to determine whether or not the sentence would appear in the output summary. The authors collected their dataset from news websites, including 20 manually summarized articles. The experimental results indicated that the proposed machine learning approach overcame the other approaches in terms of precision, recall, and F-score.

  Several researchers have used optimization algorithms for text summarization purposes. The extractive Arabic text summarization approach proposed by Al-Abdallah and Al-Taani [26] used the Firefly algorithm. The proposed approach consisted of four main steps: (1) Text pre-processing, including segmentation, tokenization, stop word removal, and stemming. (2) Calculating similarity scores using the structural feature of a sentence, including title similarity, sentence length, sentence location, and term TF-IDF weight. (3) Building a graph of candidate solutions, where vertices represent

the sentences in the original document, and the edges represent the similarity between sentences. (4) Using the Firefly algorithm to select sentences included in the summary. The proposed approach was evaluated on the EASC corpus in terms of the Recall-Oriented Understudy for Gisting Evaluation (ROUGE) metrics. Another hybrid approach proposed by Al-Radaideh et al. [27] combined domain knowledge, statistical features, and genetic algorithms to extract the essential parts of political documents. The proposed approach consisted of three main steps: document pre-processing, sentence scoring, and summary generation. The pre-processing step included the segmentation and tokenization of sentences, removed stop words, and extracted domain keywords, part-of-speech tagging, and stemming. After that, in the sentence scoring step, every sentence in the original document was assessed to determine its importance. The score of the sentence was based on several features, such as the presence of domain-specific keywords in the sentence, the frequency of words, the sentence length, the sentence position, and others. In the third step, the final summary was produced using sentence scores, cosine similarity, and genetic algorithms. The experiments were performed on two corpora: KALIMAT and EASC. The results of the proposed approach were compared to another three state-of-the-art approaches in terms of ROUGE metrics.

As shown in Table 1, much of the work reviewed concentrated on the use of extractive text summarization rather than abstractive text summarization. However, abstractive text summarization is more challenging than extractive text summarization since it refers to a new version of the original text, while extractive text summarization aims to extract the most important sentences from the original text. In the case of Arabic, fewer works on abstractive text summarization have been proposed. Al-Saleh and Menai [5] reported the following in their survey published in 2015: 'To the best of our knowledge, there exists no Arabic summarization system that can generate abstractive summaries'. After that, a very limited amount of research was proposed in this field. Most Arabic summarization approaches used graph theory and machine learning. Furthermore, hybrid approaches were proposed more frequently in Arabic rather than English. In addition, the majority of the articles reviewed were used to extract text from single documents instead of multi-documents. The EASC corpus was the most commonly used dataset for evaluating Arabic text summarization. Finally, ROUGE was the metrical assessment tool that was used the most to evaluate text summarization approaches.

**Table 1.** Recently proposed Arabic text summarization approaches.

| Ref | Year | Technique | Number of Documents | Summarization Type | Dataset | Evaluation Metrics |
|-----|------|-----------|---------------------|--------------------|---------|--------------------|
| [25] | 2015 | Machine Learning | Single documents | Extractive | 20 documents | precision, recall, F-score |
| [15] | 2017 | Graph-based | Single documents | Extractive, Abstractive | TALAA-ASC corpus | precision, recall, F-score, ROUGE |
| [27] | 2018 | Genetic algorithms | Single documents | Extractive | KALIMAT, EASC | ROUGE |
| [17] | 2018 | Ranking and rule-based | Single documents | Extractive, Abstractive | 150 documents | precision, recall, F-score, ROUGE-N |
| [23] | 2019 | Auto-encoder | Single documents | Extractive | EASC and 42 documents | ROUGE |
| [24] | 2019 | Score-based and Machine Learning | Single documents | Extractive | EASC | precision, recall, F-score, ROUGE |
| [18] | 2019 | Graph-based | Single documents | Extractive | EASC | precision, recall, F-score, ROUGE |
| [26] | 2019 | Graph-based and Firefly algorithm | Single documents | Extractive | EASC | precision, recall, F-score |
| [19] | 2020 | Graph-based | Single documents | Extractive | EASC | precision, recall, F-score |

### 3. SemG-TS: Semantic Graph-Based Text Summarization

The overall structure of the proposed abstractive single-statement Arabic text summarization model is shown in Figure 1. It starts with the representation of the original text as a semantic graph based on the proposed approach in [28], which took into account the characteristics of the Arabic language. Then, the graph embedding method is used to produce structural information from the semantic graph based on the SemanticGraph2Vec graph embedding approach proposed in [29]. The Arabic language semantic features that are stored in the semantic graph direct the semantic walks to generate appropriate Arabic-language vectors. After that, the output vectors are transferred to the deep neural network (NN) to produce the final text summary.

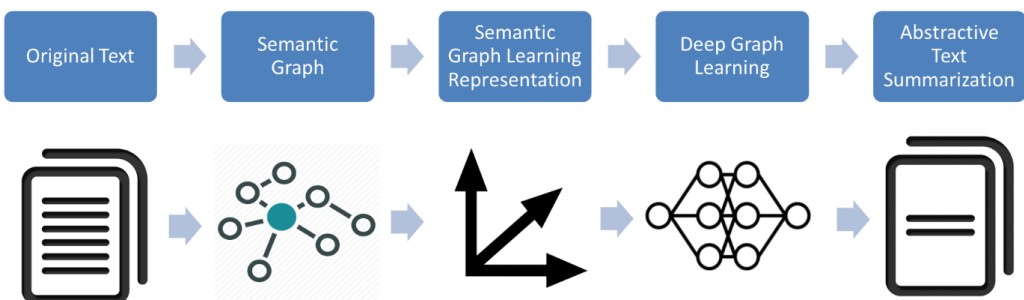

**Figure 1.** Proposed framework.

The SemG-TS consists of the following steps:

1. Data Representation: The semantic presentation of the original text becomes a main step in the proposed model, as the produced text summary focuses on retaining the main ideas of the original text. It is common knowledge that the semantic representation of the text is susceptible to the text language represented; thus, a semantic Arabic text representation graph is used in SemG-TS to consider the characteristics of the Arabic language during the construction stage of the semantic graph. A rooted acyclic semantic graph is used to represent the semantic relationships between words in the original text [28]. To the best of our knowledge, Etaiwi and Awajan [28] presented the only graph-based text representation method for Arabic language that considered the Arabic language characteristics during the text representation. Each word in the original text is represented as a vertex in the semantic graph, and the semantic relationships between the words are represented as semantic edges. Based on a set of Arabic language resources, tools, and concepts such as Arabic dependency relations, Arabic part-of-speech tags, extraction of Arabic name entities, Arabic language patterns, and predefined linguistic rules of the Arabic language, the semantic relations are extracted. Three steps comprise the building of the semantic graph: (1) Identify the word's relationships that are interdependent in the original text. (2) Use Arabic language resources to ascertain possible relationships between words (e.g., POS taggers). (3) Establish the semantic relationships between words and produce the final semantic graph in accordance with predetermined rules. A sample of the semantic graph representation is shown in Figures A1 and A2, which are listed in the Appendix A.

2. Graph Embedding: A semantic graph embedding technique is used in the graph embedding stage to preserve the semantic relationships during the graph embedding. The primary objective of graph embedding is to learn the low-dimensional representations of a graph or any of its components, such as vertices and edges, while preserving the graph's structure. However, in the application of text summarization, preserving semantic relationships becomes an important factor that can affect the final quality of the summary. During the semantic graph embedding, the SemanticGraph2Vec model referred to in [29] is used at this stage to consider the semantic relationships between words. The SemanticGraph2Vec model is a random walk-based technique

that explores the semantic graphs according to semantic priorities. The semantic relationships in SemanticGraph2Vec are dynamically sorted by frequency in accordance with their appearance in the text. As a result, the semantic relationship that is employed the most frequently will be given top priority. For instance, because it appears in texts the most, the "subject" relationship gets the lowest rank (highest priority). Finally, the high-priority relationships are more likely to be included in the walks generated.

3. Abstractive text summarization: The deep NN model is applied at this phase since deep learning outperforms other machine learning techniques in a number of NLP applications, including named entity recognition and machine translation [30,31]. The main goal of this stage is to learn the low-dimensional vectors generated from the previous stage in order to generate the text summary. The efficiency of deep learning is significantly improved by the use of the mechanisms of sequence-to-sequence learning and attention. The efficiency of statistical learning and local representations of words and phrases is evaluated in machine translation by the use of distributed word embedding in deep learning [31]. These mechanisms can also be extended to other NLP activities such as text summarization and question answering. Therefore, in the proposed model, sequence-to-sequence learning and attention mechanism were applied.

## 4. Dataset

For the Arabic language, there are no high-quality datasets that could be used to simultaneously compare the researchers' work. Most researchers in the Arabic language translated the English datasets into Arabic to validate their work, such as [32]. Few Arabic datasets are available for text summarization, such as KALIMAT [33], EASC [34], and Arabic Gigaword [35]. However, all these resources suffer from several limitations, such as the dataset's size, the dataset orientation, and the limited abstraction. For example, the EASC dataset contains only 153 documents, while the KALIMAT dataset was designed for extractive text summarization. Because academics prefer to gather their own information, the lack of Arabic standard datasets has made the evaluation process more challenging and even subjective in some instances [5]. Summary evaluation is a difficult task because there is no specific ideal summary for a given text.

Due to the limitations stated, there is a need for a large dataset for Arabic text summarization. High-quality databases with a small amount of grammatical errors may be found widely on reliable websites such as AlJazeera.net and CNN-Arabic news. Such websites contain thousands of well-written Arabic articles with single-line summaries (title and highlights). For our experiments, articles were collected from the AlJazeera.net website to build our dataset for abstractive text summarization. To evaluate the proposed abstractive single-statement Arabic text summarizer, a total of 8385 documents are used. The dataset consisted of 3,419,057 words, with an average of 5.5 characters per word. The total number of paragraphs was 16,770, with an average of 204 words in each paragraph. The titles of the articles were considered their summaries.

The articles were selected on the basis of a predefined list of keywords. The articles that were published in the last five years were considered. Keywords were carefully chosen so that they would cover a variety of different categories of articles (for example, political, sport, economy, and art). The keywords are listed in Table 2.

**Table 2.** Keywords used to build the dataset.

| Keyword | Number of Articles |
|---|---|
| "مصر" (Egypt) | 2492 |
| "سلام" (Peace) | 2299 |
| "عرب" (Arab) | 2202 |
| "اوروبا" (Europe) | 1000 |
| "رئيس" (President) | 392 |
| Total | 8385 |

## 5. Experiments and Evaluation

In order to evaluate the proposed SemG-TS model, the experiments were divided into three main parts: building the semantic graph, embedding the semantic graph, and applying deep learning for abstractive single-statement Arabic text summarization. In the first two parts, a standalone workstation with Intel Xeon Silver 4114 2.20 HHz CPU, 64 GB RAM, and Nvidia Quadro P5000 was used. In the third part, a standalone workstation with 64 GB RAM, Dual Intel Xeon E5-2620v4 CPU clocked at 2.10 GHz, and Nvidia GTX 1080 was used.

- Building the semantic graph: The first part of the experiments aimed to represent the given dataset as a semantic graph. As mentioned in [28], the process of building the semantic graph consists of two main steps:

  1. Identify the possible relationships between the words. The Farasa Segmenter is used in this step to break up the original sentence into words. In order to find subjects, objects, and adjective relations, the Farasa Part-of-Speech (POS) tagger is used. The Farasa Named entity recognizer is used to extract person and location name entities. The word's root is taken out using the Tashaphyne Arabic Light Stemmer. The original word is then compared to its root to identify the pattern of the original word.
  2. Apply predefined rules to identify the semantic relationship between words and to build the final semantic graph.

- Embedding the semantic graph: The second part of the experiments attempted to embed the semantic graph in low-dimensional vectors, in which each vertex was interpreted as a vector. SemanticGraph2vec, a customized random-walk based approach, was used to explore the semantic graph. Semantic walks were derived by the priority semantic relationship. Then, the vertices' representation was learned by optimizing the semantic neighborhood objective with the use of Stochastic Gradient Descent (SGD) with negative sampling.

- Applying deep learning for abstractive Arabic text summarization: In this part, three sets of experiments were performed. Abstractive Arabic text summarization models were trained separately on the dataset mentioned above. Seven-fold cross-validation was performed to determine the efficiency of the summarization. Essentially, the dataset was divided randomly into seven subsets of equal size. The model was trained on 6/7 of the dataset and tested on the remaining subset. Evaluation measures were stated as an average over the seven-fold validation span. The first experiment used SemanticGraph2Vec as a graph embedding method, while the second and third experiments used two versions of Word2vec. The deep learning network used in the experiments consisted of Long Short-Term Memory (LSTM) in the Encoder, LSTM BasicDecoder for training, and BeamSearchDecoder for inference. BahdanauAttention with weight normalization was used as an attention mechanism. The network had the following parameters: two hidden layers, 200 hidden units, beam width of 10, embedding size of 128, the total number of epochs was 30, learning rate of 0.005, batch

size of 128, and a keep probability of 0.80. These parameters were selected on the basis of the following sensitivity analysis.

### 5.1. Sensitivity Analysis

In a separate series of experiments, the sensitivity of the deep learning network parameters was evaluated and tested. The purpose of the sensitivity analysis was to assess the efficiency of the deep NN with respect to the parameter being examined. In the experiments, the dataset was split randomly into two parts: the training dataset containing 90% of the data and the testing dataset containing 10% of the data. The loss value was considered an evaluation measure. Several values for each parameter were examined in a variety of experiments in which all other parameters were fixed. The sensitivity analysis included the following parameters:

- Learning Rate: Three separate experiments were performed to determine the most effective learning rate. Other parameters were set, as shown in Table 3. Three different learning rate values were examined: 0.010, 0.005, and 0.001. The loss values shown in Table 4 indicate that the lower loss value was obtained using a learning rate of 0.005.

**Table 3.** Experiment parameters for the learning rate sensitivity analysis.

| Parameter Name | Parameter Value |
|---|---|
| Number of units | 200 |
| Number of hidden layers | 2 |
| Beam width | 10 |
| Embedding size | 128 |
| Number of epochs | 10 |
| Batch size | 128 |
| Keep probability | 0.80 |

**Table 4.** Sensitivity analysis results for learning rate (loss value).

| Epoch Number | Learning Rate = 0.010 | Learning Rate = 0.005 | Learning Rate = 0.001 |
|---|---|---|---|
| 1 | 65.19 | 63.55 | 63.79 |
| 2 | 55.09 | 51.83 | 56.79 |
| 3 | 53.00 | 48.12 | 53.65 |
| 4 | 47.70 | 42.82 | 51.76 |
| 5 | 45.60 | 38.00 | 48.51 |
| 6 | 42.52 | 35.27 | 44.72 |
| 7 | 37.25 | 30.44 | 44.80 |
| 8 | 34.44 | 32.48 | 40.47 |
| 9 | 31.73 | 23.16 | 35.11 |
| 10 | 28.14 | 19.96 | 31.40 |

- Beam width: Three separate experiments were performed to determine the most effective beam width. Other parameters were set, as shown in Table 5. Three different beam width values were examined: 5, 10, and 15. The loss values shown in Table 6 indicate that a lower loss value was obtained using a beam width of 10.

**Table 5.** Experiment parameters for beam width sensitivity analysis.

| Parameter Name | Parameter Value |
|---|---|
| Number of units | 200 |
| Number of hidden layers | 2 |
| Embedding size | 128 |
| Number of epochs | 10 |
| Learning rate | 0.005 |
| Batch size | 128 |
| Keep probability | 0.80 |

**Table 6.** Sensitivity analysis results for beam width (loss value).

| Epoch Number | Beam Width = 5 | Beam Width = 10 | Beam Width = 15 |
|---|---|---|---|
| 1 | 62.87 | 63.55 | 64.17 |
| 2 | 52.38 | 51.83 | 53.32 |
| 3 | 49.38 | 48.12 | 48.78 |
| 4 | 43.18 | 42.82 | 45.36 |
| 5 | 41.95 | 38.00 | 41.01 |
| 6 | 39.19 | 35.27 | 38.05 |
| 7 | 33.89 | 30.44 | 31.45 |
| 8 | 30.01 | 32.48 | 28.11 |
| 9 | 29.19 | 23.16 | 24.70 |
| 10 | 25.07 | 19.96 | 21.11 |

- Batch size: Three separate experiments were performed to determine the most effective batch size. Other experiment parameters were set, as shown in Table 7. Three different batch size values were examined: 128, 64, and 32. The loss values shown in Figure 2 indicate that the lower loss value was obtained using a batch size of 32.

**Table 7.** Experiment parameters for batch size sensitivity analysis.

| Parameter Name | Parameter Value |
|---|---|
| Number of units | 200 |
| Number of hidden layers | 2 |
| Beam width | 10 |
| Embedding size | 128 |
| Number of epochs | 20 |
| Learning rate | 0.005 |
| Keep probability | 0.80 |

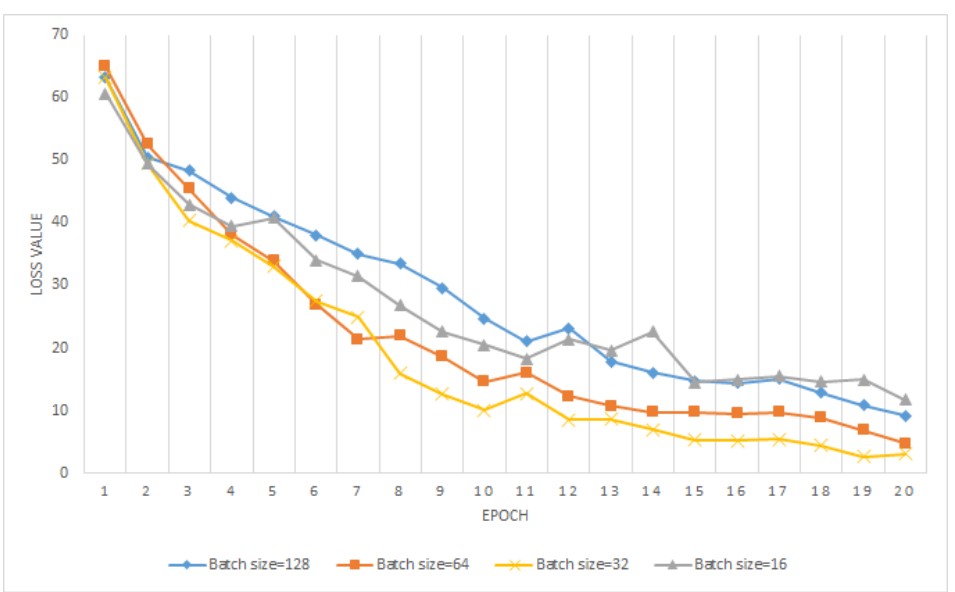

**Figure 2.** Sensitivity analysis results for batch size on the testing dataset.

- Number of units: Three separate experiments were performed to determine the most effective number of units in the deep neural network. Other experiment parameters were set, as shown in Table 8. Three different values of the number of units were examined: 100, 200, and 300. The loss values shown in Figure 3 indicate that the lower loss value was obtained using 200 units.

**Table 8.** Experiment parameters for number of units sensitivity analysis.

| Parameter Name | Parameter Value |
| --- | --- |
| Number of hidden layers | 2 |
| Beam width | 10 |
| Embedding size | 128 |
| Number of epochs | 20 |
| Learning rate | 0.005 |
| Batch size | 32 |
| Keep probability | 0.80 |

- Number of epochs: Three separate experiments were performed to determine the most effective number of epochs in the deep neural network. Other experiment parameters were set, as shown in Table 9. Three different values of the number of epoch were examined: 10, 20, and 30. The loss values shown in Figure 4 indicate that the loss value continued to decrease after 10 and 20 epochs, whereas it was almost constant after 30 epochs. This means that the deep neural network stopped learning after 30 epochs, and any additional epochs did not increase learning. The loss value did not decrease. Therefore, 30 epochs of experiments resulted in better outcome as the network continued to learn until the 26th epoch.

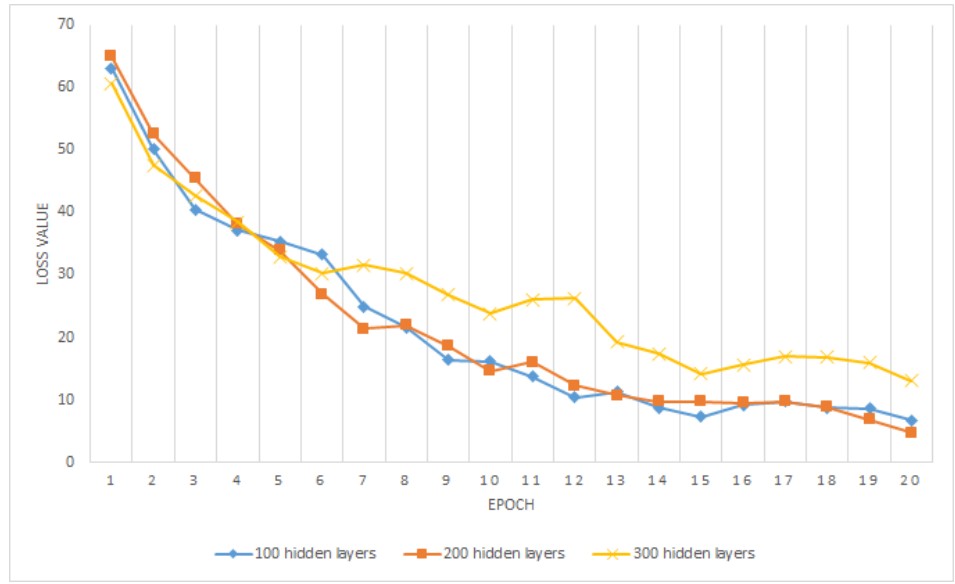

**Figure 3.** Sensitivity analysis results for number of units on the testing dataset.

**Table 9.** Experimental parameters for number of epochs sensitivity analysis.

| Parameter Name | Parameter Value |
| --- | --- |
| Number of units | 200 |
| Number of hidden layers | 2 |
| Beam width | 10 |
| Embedding size | 128 |
| Learning rate | 0.005 |
| Batch size | 32 |
| Keep probability | 0.80 |

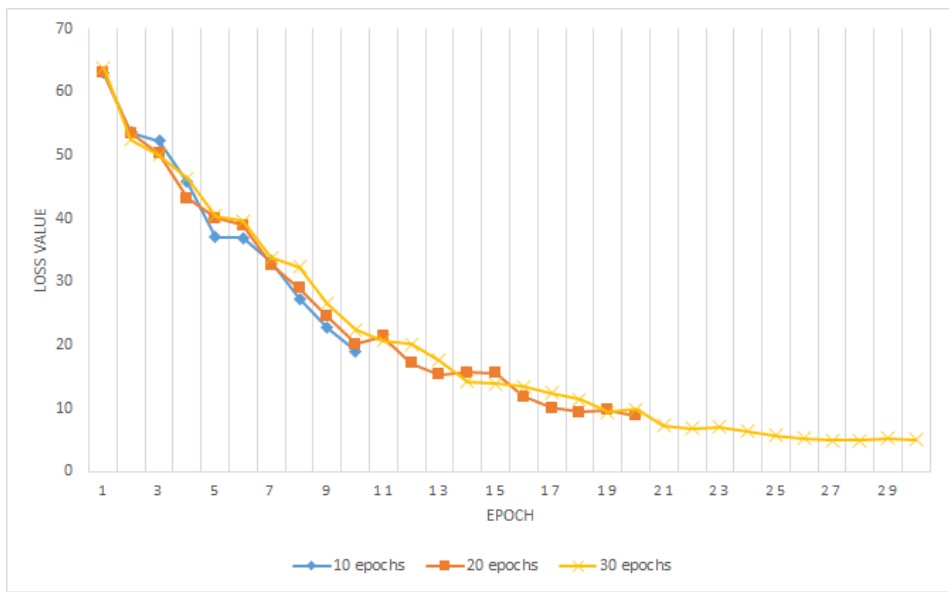

**Figure 4.** Sensitivity analysis results for number of epochs.

*5.2. Evaluation*

The results of all experiments are summarized in this section. In addition, the findings collected are discussed and used to compare the proposed SemG-TS (using semantic-Graph2Vec) with the baseline word embedding model (word2vec) in terms of their ability to enhance the quality of the abstractive Arabic text summarization. Two different types of evaluation were used: automatic evaluation and manual evaluation. The ROUGE evaluation measure was used to perform an automatic evaluation of the produced summary.

A standard automatic evaluation of summaries called ROUGE was proposed by Lin et al. [36]. It compares the produced summary (typically by the proposed models) against a set of reference summaries (typically produced by humans). ROUGE is measured based on the similarity of n-grams (an n-gram is a subsequence of *n* words) [37]. Several variants of ROUGE are used to evaluate text summarization models, such as ROUGE-N, ROUGE-L, and ROUGE-S. ROUGE-1, for instance, addresses the similarity of the unigrams in the final summary and the reference summary. The similarity of the bigrams in the produced summary and the reference summary is referred to as ROUGE-2. ROUGE-N, in general, evaluates how comparable the produced summary is to the reference summary in terms of unigram, bigram, trigram, and higher order n-grams.

**Examples 1.** *Consider the following system summary and reference summary:*

*System summary:* "جلس العصفور فوق غصن الشجرة" *(The bird sat on the branch of the tree).*

*Reference summary:* "جلس العصفور فوق الشجرة" *(The bird sat atop the tree).*

*When considering individual words, the number of overlapping words between the system summary and the reference summary is four. The recall is defined as follows:*

$$Recall = \frac{number\,of\,overlapping\,words}{total\,number\,of\,words\,in\,the\,reference\,summary} = \frac{4}{4} \tag{1}$$

*However, four out of five words in the system summary are needed or relevant. Thus, the precision is determined as follows:*

$$Precision = \frac{number\,of\,overlapping\,words}{total\,number\,of\,words\,in\,the\,system\,summary} = \frac{4}{5} \tag{2}$$

In the automatic evaluation in this paper, ROUGE-1 was considered.

Furthermore, manual evaluations were used to evaluate the produced summary by human experts. Since few works have been proposed in the field of abstractive Arabic text summarization, the experiments aimed to compare the proposed approach results with the original word2vec word embedding. The following subsections go through both evaluations in detail.

The state-of-the-art word2vec embedding model is used to compare the performance of the proposed semantic graph embedding in creating a relevant and high-quality summary. Word2vec was introduced by the Google Research Team in 2013 [38]. It is a two-layer NN that processes text and "vectorizes" words. The word2vec input is a text corpus, and the output is a set of vectors that represent the words in that corpus. The purpose of word2vec is to group vectors of similar words in vector-space. That is, it mathematically detects similarities between words. The word2vec output is a vocabulary in which each word has an attached vector that can be fed into a deep-learning network to detect the relationship between the represented words [38]. Representing words using vectors is very important for most NLP applications [39]. Thus, word2vec is used in many applications, such as sentiment analysis [40] and plagiarism detection [41].

### 5.2.1. The Automatic Evaluation

The performance of the proposed SemG-TS model is automatically compared with the performance of two versions of word2vec on the testing dataset detailed in Section 4. The first version was trained on the above dataset in order to produce the initial vectors for the text summarization, and the second version had uniform random initial vectors assigned to each word for the text summarization. The ROUGE evaluation measure was used. The three models were evaluated on the same dataset above, and the results are shown in Table 10. Clearly, the SemG-TS model, on average, surpassed the two word2vec models in all the evaluation measures. In other words, the best results of the SemG-TS model were 15.8%, 29.5%, and 21.4% better than the best version of word2vec (random-based) in terms of precision, recall, and F-measure, respectively.

**Table 10.** Automatic evaluation using ROUGE.

|  | Word2Vec (Pretrained) | Word2Vec (Random) | SemG-TS |
|---|---|---|---|
| Precision | $2.26 \times 10^{-2}$ | $3.92 \times 10^{-2}$ | $4.54 \times 10^{-2}$ |
| Recall | $2.42 \times 10^{-2}$ | $3.92 \times 10^{-2}$ | $5.08 \times 10^{-2}$ |
| F-measure | $2.30 \times 10^{-2}$ | $3.87 \times 10^{-2}$ | $4.70 \times 10^{-2}$ |

Although the three models had a low ROUGE performance, it can be noted that SemG-TS surpassed the baseline word embedding model (word2vec) in both versions, which satisfies the main goal of the experiments. These results might be improved by a number of methods, such as: (1) Use more data for training. This is merely an effect of the amount of data in any deep learning model [42,43]. (2) Improve the quality of the dependency parser used to create a semantic graph so it can have more accurate dependency relationships between words. (3) Use a collection of more semantic relationships that enrich the semantic representation.

The loss values of the three experiments are shown in Figure 5. Loss values will continue to decrease with more epochs, which will decrease until they are stable. Therefore, the loss value was almost constant in the last three epochs.

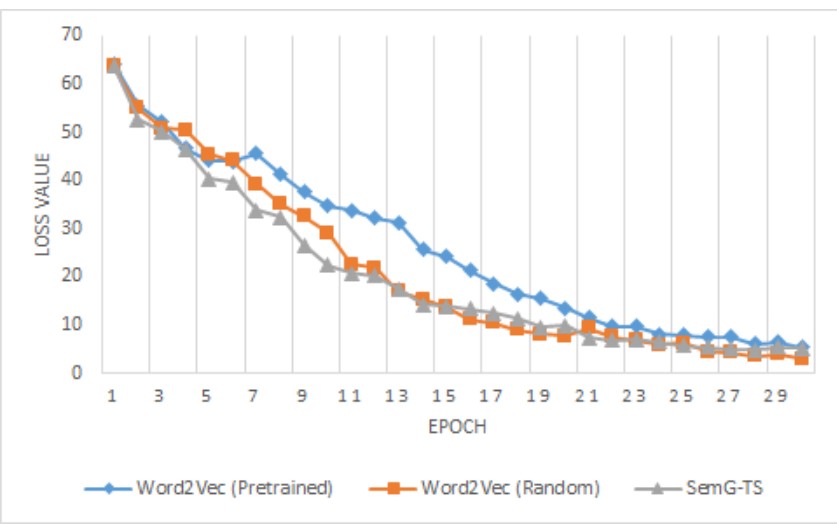

**Figure 5.** Loss values for experiment with 30 epochs.

5.2.2. The Manual Evaluation

The ROUGE evaluation measures compute the frequency of the overlapping n-grams of the summary produced and the reference summary; thus, the abstractive text summary may not contain the same words that were used in the reference summary, which makes the ROUGE evaluation method irrelevant for this type of application. The proposed abstractive text summarization model was therefore evaluated manually by human experts. The evaluators were required to assess both the proposed summary and the summary produced by the baseline algorithm. The original text and the two summaries were provided on the main screen of a customized application, and four main questions maintained the relevance, similarity, readability, and overall satisfaction of the evaluated summary.

A simple desktop application was built for the assessment of an evaluator, as the correct way to know whether the proposed summary is relevant and appropriate is to receive feedback directly from a human being. A sample screenshot of the evaluation screen is shown in Figure 6, and more samples are listed in the Appendix B. Element number 1 refers to the article ID and the article title, element number 2 refers to the article body. Suggested summaries are shown in element number 3. Finally, the evaluation questions are set out in element 4.

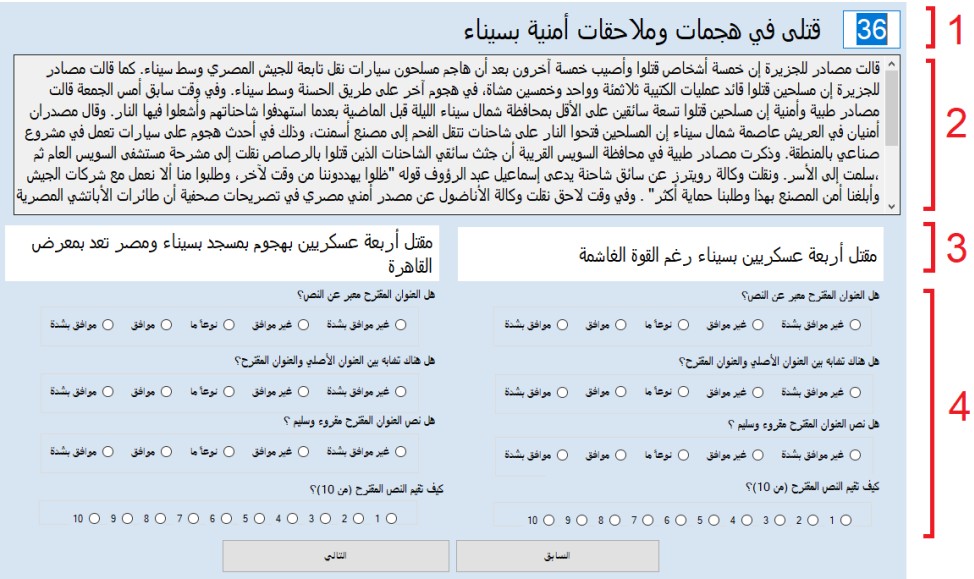

**Figure 6.** Screenshot of manual assessment application.

Every text summarization model was set to produce its summary for the article displayed. The order in which the two summaries appeared below the article was randomized. As a result, the evaluator did not know which model the summary being shown was referring to.

Next to the article body, four assessment questions were listed in order to get feedback about the relevancy of the summary, similarity between the suggested summary and the original summary, the suggested summary's readability and quality, and overall satisfaction. Evaluators were also provided with five radio buttons to indicate the relevancy, similarity, and readability of the suggested summary: (a) I totally agree, (b) I agree, (c) Maybe, (d) I disagree, and (e) I totally disagree. However, a scale of 1–10 was used to indicate the overall satisfaction of the evaluator with the suggested summary, with a higher value indicating more satisfaction. The questions on the assessment screen were the following:

1.  Does the proposed summary express the text? This question refers to the relevance of the proposed summary to the original text. It should consider the similarity of the domain, the semantic similarity, and the principal similarity of the main ideas expressed in the article.
2.  Is the proposed summary similar to the title of the article? This question measures the similarity of the title of the article and the suggested summary in terms of domain, content, and keyword similarities.
3.  Is the proposed summary readable and logical? This question evaluates the consistency of the suggested summary in terms of linguistic integrity and logical validity.
4.  What is your overall level of satisfaction with the suggested summary? This question measures the overall satisfaction of the evaluator with the suggested summary.

Evaluators were selected based on several conditions, such as: The mother tongue of the evaluator should be the Arabic language, the evaluator should hold a post-graduate degree, and also should be from different scientific backgrounds. In addition, 605 articles were chosen randomly from the dataset, and three evaluators submitted their feedback on both suggested summaries.

Figure 7 shows the results of the frequency of summary relevancy, similarity, readability, and overall satisfaction. Looking at these raw data, SemG-TS and word2vec are similar to each other in terms of relevancy and how near their summaries are to the reference summary. However, in terms of readability and overall satisfaction, SemG-TS has more fragmented data at all levels than word2vec. In comparison, word2vec has more rank-1 (lower rank) than SemG-TS.

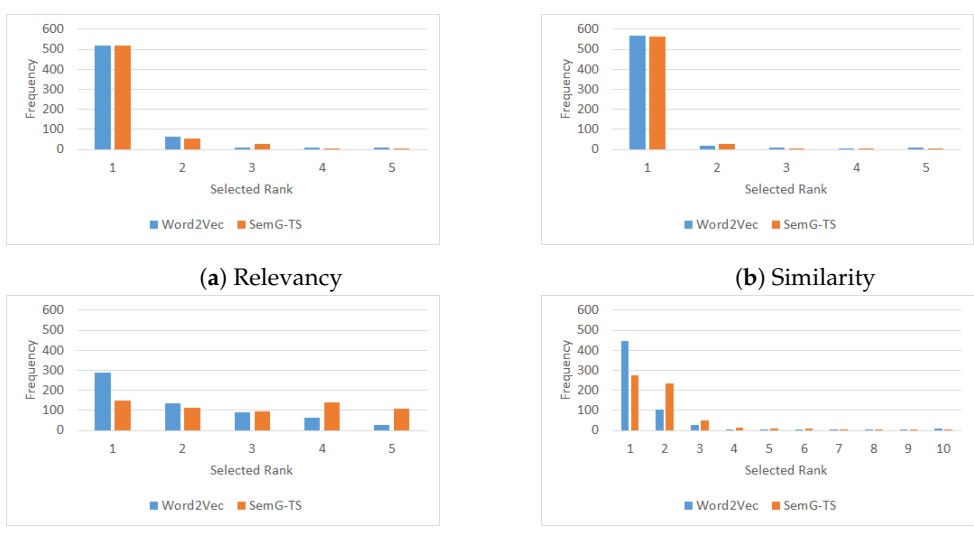

(**a**) Relevancy

(**b**) Similarity

(**c**) Readability

(**d**) Overall Satisfaction

**Figure 7.** The frequencies of the evaluators' responses.

Table 11 summarizes the basic statistics of the summary assessment from the evaluators. The mean scores in relevancy and similarity are slightly higher for the SemG-TS model, while the mean scores in readability and overall satisfaction are significantly higher for the SemG-TS model. However, considering the scales of the ratings, the SemG-TS manages to increase readability significantly with marginal sacrifices in relevancy.

Paired-sample t-tests were conducted to compare the means between the summary produced by the proposed model and the summary produced by the baseline model. There was a significant difference in the relevancy scores of SemG-TS and word2vec; $t(604) = -1.64$, $\alpha = 0.05$. There was also a significant difference in the readability scores of SemG-TS and word2vec; $t(604) = -10.47$, $\alpha = 0.05$. According to the overall satisfaction score, there was also a significant difference in the overall satisfaction scores for SemG-TS and word2vec; $t(604) = 4.72$, $\alpha = 0.05$. However, there was no significant difference in similarity, with $t(604) = -0.35$, $\alpha = 0.05$.

As the distribution of data was chi-squared, as seen in Figure 7, the chi-squared test was used to evaluate the likely of the frequency of the evaluation results for SemG-TS and word2vec to be substantially different. The chi-squared test showed that there was a significant difference between the two models in terms of readability and overall satisfaction, with 978.18 and 701.3 chi-square values ($\alpha = 0.05$) for SemG-TS and word2vec, respectively. Conversely, the chi-squared test showed no significant difference between the two models in terms of relevancy and similarity, because the chi-square values were 196.58 and 157.68 ($\alpha = 0.05$), respectively, for SemG-TS and word2vec.

**Table 11.** Basic statistics of the manual evaluation.

| | Relevancy | | Similarity | |
| --- | --- | --- | --- | --- |
| | SemG-TS | Word2Vec | SemG-TS | Word2Vec |
| Count | 605 | 605 | 605 | 605 |
| Mean | 1.23 | 1.23 | 1.11 | 1.14 |
| STD | 0.65 | 0.70 | 0.48 | 0.62 |
| Percentage | 4.7% | 4.7% | 2.1% | 2.8% |
| | Readability | | Overall Satisfaction | |
| | SemG-TS | Word2Vec | SemG-TS | Word2Vec |
| Count | 605 | 605 | 605 | 605 |
| Mean | 2.91 | 2.01 | 1.89 | 1.55 |
| STD | 1.45 | 1.20 | 1.35 | 1.44 |
| Percentage | 38.2% | 20.3% | 18.9% | 15.5% |

## 6. Conclusions

In this research, SemG-TS, an abstractive single-statement Arabic text summarization model is proposed. The proposed model consists of three main steps: the construction of a semantic graph, the embedding of a semantic graph, and the production of a final summary. The semantic random-walk-based approach, called SemanticGraph2vec, was applied in the embedding step, and then deep NN was used to produce the final summary. A new dataset consists of news articles that had been collected from the Al-Jazeera.Net website was used in the experiments. Two different types of evaluation were used: automated evaluation and manual evaluation. The ROUGE evaluation measure was used for automatic assessment. Conversely, a manual evaluation was carried out by human experts to measure the relevancy, similarity, readability, and overall satisfaction of the proposed summaries. The results were compared to the word embedding model, word2vec. The experimental results show that the proposed SemG-TS model surpasses word2vec in terms of ROUGE, relevancy, readability, and overall satisfaction.

**Author Contributions:** Conceptualization, W.E.; Methodology, W.E.; Project administration, A.A.; Resources, W.E.; Software, W.E.; Supervision, A.A.; Validation, W.E.; Writing—original draft, W.E.; Writing—review & editing, A.A. All authors have read and agreed to the published version of the manuscript.

**Funding:** This research received no external funding.

**Institutional Review Board Statement:** Not applicable.

**Informed Consent Statement:** Not applicable.

**Data Availability Statement:** Not applicable.

**Conflicts of Interest:** The authors declare no conflicts of interest.

## Appendix A. Semantic Graph Representation Samples

In this appendix, different examples and test cases of the used semantic graph are illustrated and discussed. The first example represents the sentence:

"اغتيال مدير هيئة العدالة والمسائلة في بغداد"

("The assassination of the director of Justice and Accountability Commission in Baghdad"). There is no verb in this sentence. It contains the location noun "بغداد" (Baghdad) and the conjunction word "و" (and). Each word in the sentence is represented as a distinct vertex, as seen in Figure A1. The location is represented by an extra concept vertex.

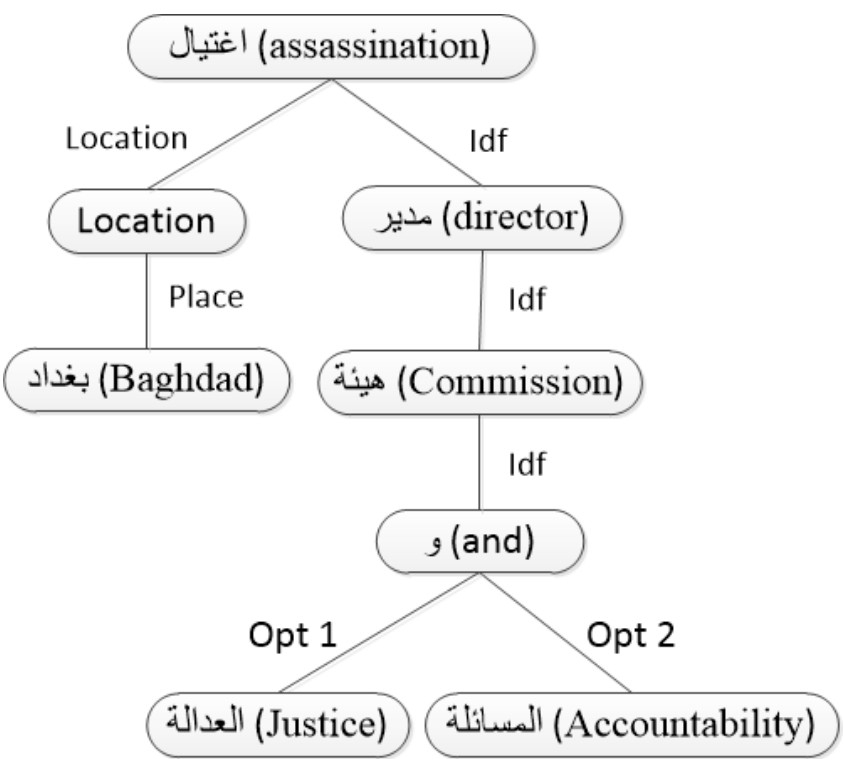

**Figure A1.** Semantic representation sample number 1.

The second example, illustrated in Figure A2, represents the sentence:

"قرر الجيش الأمريكي خفض عدد قواته في الباكستان خلال العام المقبل"

("The US military has decided to reduce the number of its troops in Pakistan during next year"). The semantic graph in this example is expanded to include vertices for the location and date/time concepts. A verb and its properties are present in the sentence (subject and object).

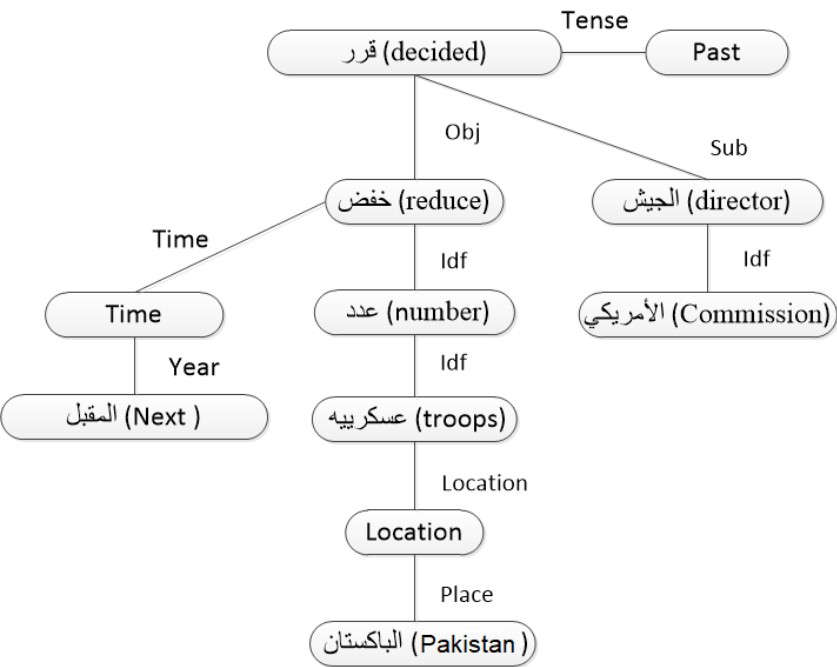

**Figure A2.** Semantic representation sample number 2.

## Appendix B. Produced Summary Samples

Examples of the summaries that were created using the proposed approach and the word2vec word embedding model are shown in this appendix. Screenshots of the manual evaluation application's home screen are shown in Figures A3–A8. The original text and the two summaries are shown in random order on each screenshot.

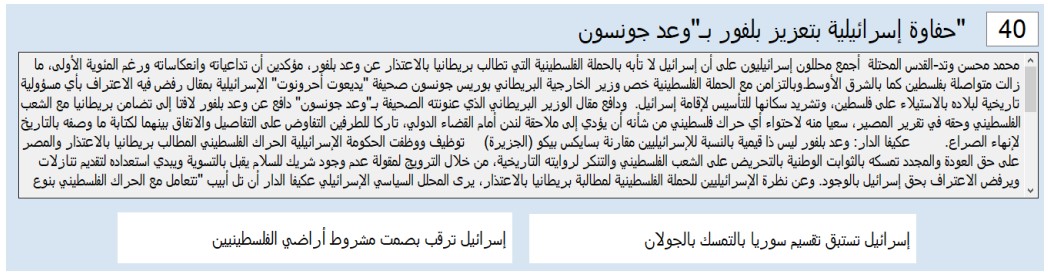

**Figure A3.** Sample number 1.

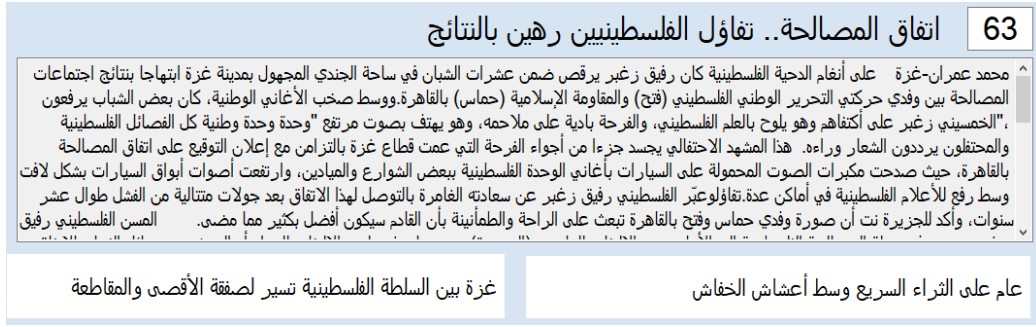

**Figure A4.** Sample number 2.

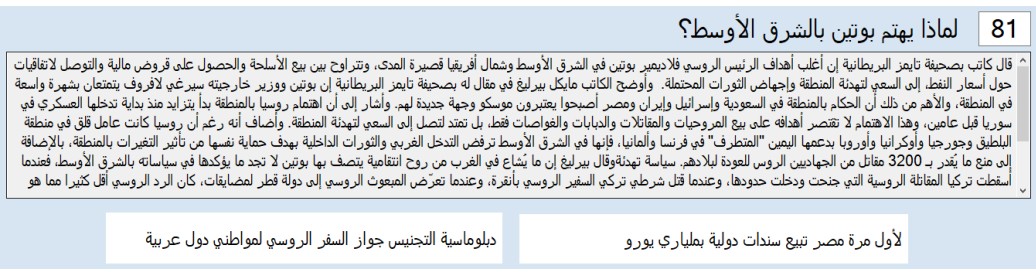

**Figure A5.** Sample number 3.

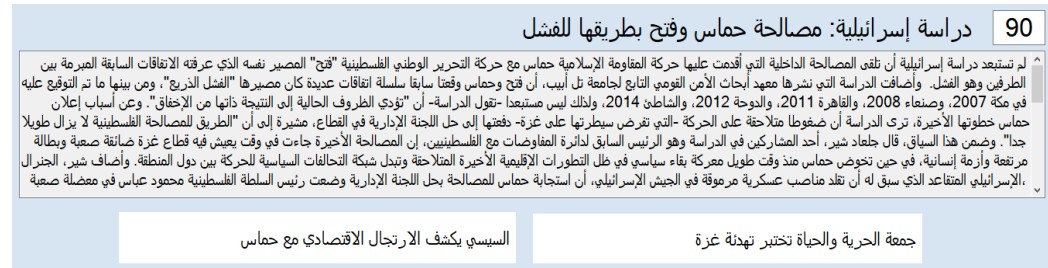

**Figure A6.** Sample number 4.

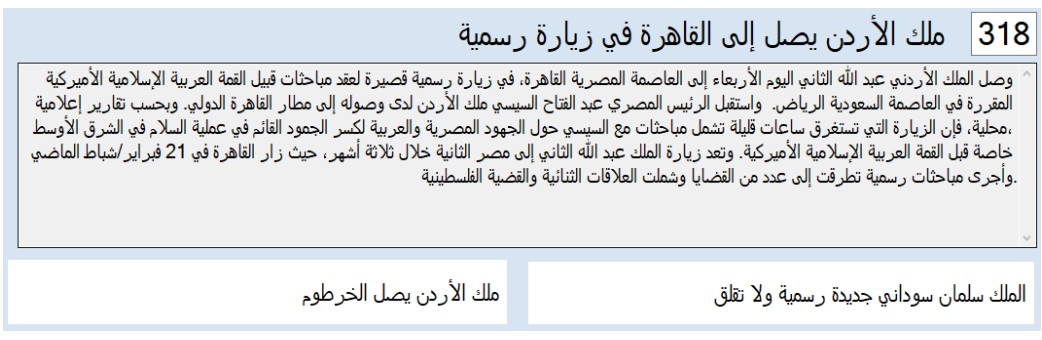

**Figure A7.** Sample number 5.

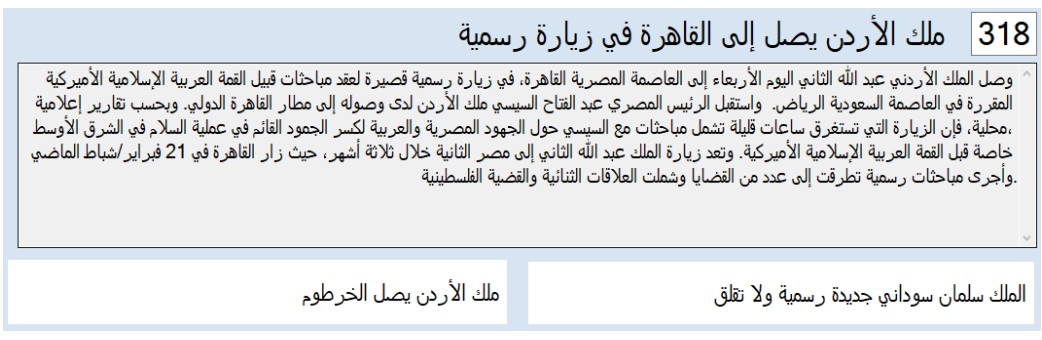

**Figure A8.** Sample number 6.

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
