# Peer review of "SemG-TS: Abstractive Arabic Text Summarization Using Semantic Graph Embedding"

_mathematics, doi:10.3390/math10183225_

Round 1
Reviewer 1 Report
Dear editor and authors,
Please see the report in the attached file.
Regards

Author Response
I would like to thank the editor and the reviewers for their invaluable comments. I have addressed each one of the comments in the attached document.

Reviewer 2 Report
The first and main problem that this paper poses is that two important foundations, namely positions [27] and [28] from bibliography, aren't described enough to understand the proposed solution. While readers can reach "Arabic text semantic representation" published in the Information Processing & Management, the contents of [28] (PhD thesis) are nearly impossible to read - how to obtain the text? In effect, section 3. SemG-TS is too brief. Some specific points are listed below, but in general, it would be useful to clearly define inputs and outputs of each of the steps.
Specifically:
1. The description in lines 243-248 should be extended with more information on how the graph generation is designed. How are the words linked to each other? Readers can only guess that to link words within one sentence, perhaps a dependency parser is used (btw what is the role of part-of-speech tags?), but how to link multiple sentences into one graph? Maybe a diagram or a figure would be useful that illustrates a fragment of a graph. I realize it can mean covering some material from [27]. Lines 311-319 are not helpful as they only list exact NLP tools.
2. Similar remarks concern the SemanticGraph2Vec covered in lines 256-260, all we know is that "it is a random walk based technique that explores the semantic graphs according to semantic priorities". What are the priorities? "the semantic relationships are ordered", but ordered according to what? How exactly the random walk is implemented to generate graph embedding? Are word embeddings used? We know that the output from this procedure is low dimensional vectors, but what do they represent? One vector represents one graph, which corresponds to one text with multiple sentences?
3. Line 263 mentions a neural network (NN) used to generate text from the low dimensional vectors / output from the previous step. Lines 335-339 indicate that the network is an LSTM. Why didn't you use pre-trained transformers, that are currently state-of-the-art in most languages for the text summarization task?
The second problem, word2vec usage is confusing. In the abstract we read: "popular baseline graph embedding technique called word2vec" <- this is not true. According to the papers by Mikolov et al and also the Wikipedia page (https://en.wikipedia.org/wiki/Word2vec) Word2vec is a group of related models that are used to produce word embeddings. It can be adapted to produce graphs as one of intermediate steps and word embeddings can be used all along, but other steps have to be done and thys explained. In other words, an explanation of how to produce graphs from text using word2vec is necessary because the original word2vec papers do not deal with graphs in any way. Line 334 indicates that word2vec vectors an input for LSTM network - but is this really a graph or maybe simply, a word embedding technique?
It is also hard to understand the second type of word2vec usage, called "random" (eg. Table 10). I could not find a good explanation.
Minor issues:
Line 22: "need for a text summarization solution. Text summarization compresses a
large volume of texts" - how about avoiding repetition and reformulating into "need for a text summarization solution to compress a large volume of texts"? It reads easier. Please take care of other such issues in the text.
Why do you introduce abstractive vs extractive summarization twice? It's done in lines 21-22 and 29-37. It seems that lines 21-22 are redundant and should be removed.
In lines 45-49 you should explain in more detail why Arabic summarization is more difficult than in English. The provided explanation "complexity of the Arabic language, both in terms of syntax and morphology" is a bit too concise.
Line 86 - SemanticGraph2Vec has to be better introduced here. No references or descriptions are provided.
Line 93 "a briefly review" -> "a brief review"
Lines 229-230: semantic features stored in the semantic graph direct the random walks ? Are the random walks really random, if controlled / directed by semantic features stored in the semantic graph?
Table 6. does not contain any information in the caption about the measure reported, please add that it is loss.
Figures 2 and 3 should include in the caption what dataset was used - was it computed on the validation set?
Line 302 should contain a word2vec citation.
Line 403 "testing dataset detailed above" - described where above? please refer to subsection number. I do not find any dataset information in section 5.2 which is directly above.
Section 5.2.1 - I do not understand several points:
- what exactly are word2vec-random? line 404 says "the second version has uniform random initial vectors". There is also Table 10 with the column "Word2vec (Random)". How can word2vec vectors be random? The training procedure using CBOW or skip-gram approach makes them non random, this is the whole point of pre-training word embeddings.
- in Table 10, I do not understand how precision, recall and F-measure are defined regarding the ROUGE, it compares an automatically produced summary against a reference. It is defined as an overlap of n-grams. How is overlap of n-grams associated with precision, etc?
- Which rouge is used, by the way? There are ROUGE-N, ROUGE-1, -2, -L and other variants.
Author Response

(The authors gave the same response as above.)
